# An Assessment of Environmental RF Noise Due to IoT Deployment

**DOI:** 10.3390/s23187899

**Published:** 2023-09-15

**Authors:** Dominique G. K. Ingala, Nelendran Pillay, Aritha Pillay

**Affiliations:** Department of Electronic and Computer Engineering, Durban University of Technology, Durban 4001, South Africa; trevorpi@dut.ac.za (N.P.); arithap@dut.ac.za (A.P.)

**Keywords:** environment RF noise, equipment RF noise, IoT, ISM bands, noise generators, RF noise survey, software-defined radio

## Abstract

The advent of the Internet of Things (IoT) has contributed to an increase in the production volume of RF-featured equipment. According to statistics from the literature, the IoT industry will soon deploy billions of products. While the concept behind these applications seems exciting, this paper sought to assess the effects the radio emissions produced by IoT products would have on the ambient radio noise levels within the unlicensed frequency bands of 433 MHz, 868 MHz, and 2.4 GHz. The study extended to three environments: industrial, urban, and suburban. This study developed an IoT noise generator (ING) device to emulate RF noise signals in the desired IoT radio transmission band. The paper presents a simplified radio noise surveying system (RNSS) for data collection of ambient radio noise from five South African candidate sites. The statistical and empirical analysis agree that the level of ambient radio noise was directly proportional to the rate of IoT radio activities. The slopes of the regression lines demonstrate that 80% of the analyzed data developed augmenting trends. Approximately 20% of the data show declining trends.

## 1. Introduction

### 1.1. Research Background

Studies conducted by [1,2] explain that manufactured technologies of the IoT industry are prone to become ubiquitous in modern civilization. The new developments in electronics and communication technology (RF semiconductors, system-on-chip micro-controllers, computing processors miniaturization) enable flexible conceptualization and rapid implementation of numerous IoT applications. Systems such as autonomous vehicles, smart cities, domotics, health, industrial IoT, smart farming, e-agriculture, smart metering, advanced tracking and surveillance, and intelligent public transportation, to name a few, are nowadays omnipresent in all sectors.

Ericsson’s Mobility Reports [3,4,5] clearly illustrate the tremendous evolution of mobile technology and the increase in mobile handhelds, base stations, mobile subscriptions, and data traffic. For instance, they predicted that 2022 would witness 29 billion connected nodes, of which 18 billion would be in the IoT domain [3]. There has been massive development in cellular IoT connections in the 4G, LTE, NB-IoT, Cat-M, and 5G bands: 4.4 billion wide-area IoT, 4.1 billion cellular IoT, and 17.8 billion short-range IoT connections. Approximately 1.9 billion 5G subscriptions are expected by the end of 2024 [5].

### 1.2. Research Aim

The paper hypothesized that the higher the environmental noise floor level, the more radio emissions increase. According to Enge [6], the availability of the free industrial, scientific, and medical (ISM) parts of the radio spectrum encouraged countless IoT applications. Consequently, any negligence towards policy and regulatory conformity may cause adverse consequences to the IoT industry and deteriorate some wireless operations. It was, therefore, essential to justify with evidence if these concerns remained applicable to IoT development characterized as low RF power applications. The main aim of this research was to statistically evaluate the proportional relationship between the level of ambient radio noise and the emissions rates generated by artificial IoT RF apparatuses. This paper examined three ISM frequencies, 433 MHz, 868 MHz, and 2.4 GHz, in industrial, urban, and suburban areas distributed over five Durban (South Africa) candidate sites. The IoT radio technologies include Wi-Fi, Bluetooth, ZigBee, Sigfox, Lora, and proprietary protocols, commonly operating under 433 MHz, 868 MHz, 915 MHz, and 2.4 GHz frequencies. The experiment had to comply with the local regulations applied to ISM band usage. Except for the 915 MHz, the Independent Communication Authority of South Africa (ICASA) approved the 433 MHz, 868 MHz, and 2.4 GHz ISM bands [3,7,8]. This research applied Universal Software Radio Peripheral (USRP) as a tool for software-defined radio (SDR), and the GNU Radio system for data collection. The study employed descriptive statistical analysis and heuristic empirical methods to provide informative insight.

### 1.3. Contributions and Novelties

This study’s primary challenge was the lack of enough IoT radio noise data already in the environments. To the best of our knowledge, this research is the first attempt to examine the direct impact of IoT radio noise in South African industrial, urban, and suburban environments. At the time of this study, the IoT Industry observed limited deployment, contrary to predictions. As a critical contribution, this paper had to innovate by initiating the design, development, and production of IoT noise generators (ING) to emulate IoT product behaviors in radio transmissions. IoT radio noise surveying is necessary in many parts of the world. Even though the experiments took place in specific locations in Durban, this paper presents replicable routines for noise generation, data collection, and analysis to provide valuable trends for other places worldwide. To handle big data, the studies in [9,10,11] introduced parallel and distributed computing tools, such as Hadoop for data storage and MapReduce, Spark, Splunk, and Skytree for data processing. Applying these methods would be contrary to the objective of this paper to provide a simplified solution. This paper documents a strategic approach to data reduction on averaging by data segmentation [12].

### 1.4. Structure

In Section 1, the introduction covered the research background, aim, discussions, and novelties. Section 2 is the literature review. Section 3 discusses the development of the RNSS. Section 4 introduces the design cycle of the ING units from concept, design, assembly, and functionality tests. Section 5 elaborates on the data collection campaign at the candidate sites and the operating procedure to record the data for this research. Section 6 relates to the methods used to prepare the data for analysis. It highlights the strategy to transform raw data into clean datasets. Section 7 applies a descriptive statistical analysis. Section 8 discusses the results of the statistical nature of the collected data. Regression lines demonstrate the direction of trends. Finally, Section 9 applies a heuristic and an empirical analysis to examine IoT operation impacts on the environment radio noise.

## 2. Literature Review

The topic of IoT has been approached several times with different interests. One of the trends focuses on architectural IoT and business opportunities. Guo et al. [13] described that recently, researchers focalized on the “thing” itself from a device-based approach to implementing specific IoT applications beneficial for humans. Qin et al. [1] discussed the IoT techniques based on data specifications with an interest in data processing and storage. They suggested that the IoT industry would be one of the accelerants of big data science. Neshenko et al. [14] highlighted IoT vulnerability issues: attacks against confidentiality, integrity, accountability, and authentication, and countermeasures. Karie et al. [15] elaborated on the security perspectives: data leakage, eavesdropping, hacking, software exploitation, and device security. All these noteworthy publications have considered that the things are already successfully operable and are located in friendly vicinities, which may not be the case; thus, the contribution of this study is to evaluate a separate issue, namely, the impact of massive deployment of IoT technologies over the ambient radio noise floor. Furthermore, we aim to provide relevant insight into the contribution of the IoT RF operations to environmental radio noise.

Spaulding [16] expected a detrimental performance from RF systems when the spectrum becomes increasingly crowded by interference and radio noise generated by various sources. It is vital to monitor the ambient electromagnetic levels continuously.

According to Weinmann and Dostert [17], a common approach for estimating the noise level is to refer to the ITU report [18]. This document provides detailed information to calculate the amplitude of ambient radio noise. Still, experts have raised concerns that this referral ITU study is now too old, as their data came from measurements conducted more than 50 years ago. Reports have questioned the essence of ITU recommendations as, after half a century, technology has advanced many changes that the electromagnetic environments have witnessed. If they raised this question in a study conducted in 2006, then it provided more reasons to work on new measurements to obtain actual data. Their paper also describes measurement techniques performed over short-wave frequencies (30 MHz) using loop antennas; however, following these approaches and utilizing equipment at much higher frequencies could not guarantee success.

Skeie and Solberg [19] argued that the same ITU referral report based its analysis and results from the data collected in environments in the United States of America. For higher accuracy, conducting the same exercise at different locations was relevant. Bradshaw [20] conducted similar studies in Australia. This was a motivation to assess ambient RF noise in South African environments.

Jejdling [5] indicated that IoT operates with different commercial communication technologies and frequency bands; however, another report [3] categorized wide-area IoT devices as low-power RF devices on unlicensed frequency bands. Research conducted by Enge [6] included survey campaigns on artificial emissions in other frequency bands: 1563.42 to 1587.42 MHz, 2025 to 2110 MHz, 2400 to 2482.50 MHz, and 23.6 to 24.0 GHz. Regarding the frequency bands of interest, this paper limited its scope to explore 433 MHz, 868 MHz, and 2.4 GHz, which may be the three most exploited ISM frequency bands in the South African IoT and M2M Industry.

The RF noise receiver system is crucial to the ambient RF noise survey equipment. This paper employed the Universal Software Radio Peripheral (USRP) and software-defined radio (SDR) system. Vasudeva et al. [21] used the same hardware platform to analyze measurements for connectivity for vehicles connected over the LTE network in urban and rural locations. They combined the USRP N210 device with LabView as an SDR system and post-processing in MATLAB; however, to produce a cost-effective RNSS, this study used GNU Radio software version 3.8.1.0 [22] as part of the SDR tool. The integration of USRP and GNU Radio has shown reliability. Additionally, GNU is a free software that contributes to achieving low-cost development.

This literature review found limited resources from previous studies on ambient RF noise or environmental RF interference in South Africa. Dunn [23] experimented with RFI measurements from the Square Kilometer Array (SKA) environment in South Africa; however, their campaign surveyed only remote and reserved locations per the SKA guidelines. In contrast, this research had focused primarily on populated agglomerations, which would possibly present different characteristics than the SKA’s uninhabited places.

The South African Radio League (SARL) is another organization concerned with monitoring the level of ambient radio noise. As reported by Groenendaal [24], SARL concluded that the noise floor in the radio spectrum was rising as the number of devices emitting radio energy increased; however, it was challenging to support this claim without quantitative data. This paper, therefore, makes an effort to address SARL’s concerns.

## 3. Radio Noise Surveying System

The study of [12] was a precursor to this current paper, and introduced the concept of this RNSS. It demonstrated that it was technically possible to survey ambient radio noise using simplified, cost-effective SDR technology. Unlike the current paper, the previous experiment only explored the “quiet” environments unimpacted by IoT emissions. The ING devices in this paper introduce IoT radio activities’ influence. Figure 1 shows the system block diagram. A USRP2 motherboard integrated with an SBX daughterboard (400 MHz to 4.4 GHz) and controlled by Gnu Radio software version 3.8.1.0 formed the hardware branch. Based on the frequency, the RNSS used three different terminal antennas. Figure 2 is the GNU Radio companion (GRC) flowgraph for the RF receiver architecture. The system extracted only the real part of the complex data. This study did not require the imaginary part. A QT sink block constantly monitored the operation underway [12].

The theory of complex envelope in RF and digital signal processing (DSP) associates noise power with the real part of a complex signal. The analysis of discrete Fourier transform (DFT) in [25] provides the complex notations for signals in the frequency domain. Equations (1) and (2) transform to the frequency domain by multiplying time domain samples by sine or cosine functions. The DSP terms call it correlating the input signal with the basis function.
(1)ReX[k]=∑i=0N−1x[i]cos(2πkiN) 
(2)ImX[k]=−∑i=0N−1x[i]sin(2πkiN) 
where x[i] is the signal in the time domain, the index i increments from 0 to N−1, and the index k from 0 to N/2. The real part is the cosine wave and the imaginary sine wave, both at the same angular frequency. The sum of the cosine and sine signals results in a cosine wave (of different amplitude and phase shift). Instead of a rectangular representation, the polar form is practical in frequency domain analysis. Equations (3) and (4) introduce two components for the polar form: the magnitude MagX[k] and the phase PhaseX[x] of the cosine waves.
(3)$$MagX\left[k\right]=\sqrt{ReX{\left[k\right]}^{2}+ImX{\left[k\right]}^{2}}\;$$
(4)PhaseX[x]=arctan(ImX[k]ReX[k])

Equations (5) and (6) rewrite Equations (3) and (4) to a rectangular form and represents the real and imaginary parts dependent on the magnitude and phase. The polar form does not use a sine wave to describe signals because a sine wave cannot have the DC component of the signal, and the sine wave is nil at 0 Hz.
(5)ReX[k]=MagX[k]cos(PhaseX[x]) 
(6)ImX[k]=MagX[k]sin(PhaseX[x]) 

The theory of signals and bandpass systems in [26] explains that the Fourier transform X(f) of an original signal x(t) includes negative and positive components symmetrically around 0 Hz. The Fourier transform is a complex signal where the even function represents the real part, and the odd function is the imaginary part. Filtering out the right-hand side of the X(f) is necessary to reconstruct the original signal x(t). As shown in Equations (7) and (8), a linear system retrieves the real part from the complex analytic signal $\dot{x}\left(t\right)$ to recover the original baseband x(t), where $\tilde{x}\left(t\right)$ is referred to as the complex envelope. $f_{0}$ denotes the frequency at the baseband (0 Hz).
(7)$$\dot{x}\left(t\right)=\tilde{x}\left(t\right)e^{j2\pi f_{0}t}$$
(8)$$x\left(t\right)=Re\left[\dot{x}\left(t\right)\right]=Re\left[\tilde{x}\left(t\right)e^{j2\pi f_{0}t}\right]$$

Equations (9) and (10) show the instantaneous envelope and instantaneous phase of x(t), where $x_{c}$ and $x_{s}$ are the quadrature components of $\tilde{x}\left(t\right)$.
(9)$$A\left(t\right)=\left|\tilde{x}\left(t\right)\right|=\sqrt{x_{c}{}^{2}+x_{s}{}^{2}}\;$$
(10)θ(t)=arctan(xsxc)

Similarly, the study in [27] elaborates that the positive frequencies of the Fourier transform relate to the Hilbert transform at a negative phase shift of −90 degrees. On the contrary, the negative frequencies relate to the Hilbert transform at +90 degrees. Equation (11) defines the positive frequencies of the analytical signal, hence s(t) and $\hat{s}\left(t\right)$ are passband real and imaginary parts, respectively. $f_{c}$ denotes the passband component at a given carrier frequency, and $\tilde{s}\left(t\right)$ is the complex envelope.
(11)$$s_{+}\left(t\right)=s\left(t\right)+j\hat{s}\left(t\right)=\tilde{s}\left(t\right)e^{j2\pi f_{c}t}$$

In Equation (12), s(t) is the real part of $s_{+}\left(t\right)$ and represents an essential concept of the complex envelope to retrieve the passband signal as:(12)$$s\left(t\right)=\left|s_{+}\left(t\right)\right|=Re\left[\tilde{s}\left(t\right)e^{j2\pi f_{c}t}\right]\;$$

The same logic applies to noise voltage developed in [28,29]. As an instantaneous or time-varying quantity, the noise voltage v(t) in Equation (13) defines the real part of the noise voltage complex envelope $\hat{v}$ about the carrier frequency $f_{c}$.
(13)$$v\left(t\right)=R_{e}\left\{\hat{v}\left(t\right)e^{j2\pi f_{c}t}\right\}\;$$

Equation (14) describes the instantaneous noise power obtained by squaring the absolute value of the time-varying complex envelope.
(14)$$w={\left|\hat{v}\left(t\right)\right|}^{2}\;$$

## 4. IoT Radio Noise Generation

Although studies have predicted massive deployment of the IoT industry, at the time of this writing, South African environments are still not populated much with IoT operations at a large scale. It was, therefore, necessary to develop a mechanism for generating RF noise emulating the IoT radio transmissions.

Qin et al. [1] highlighted the IoT’s complex background in data generation. By operational definition, IoT connects every object: human clothing, robots, home appliances, mobile phones, and more. In 2012, it was recorded that about 2.5 quintillion (2.5 × 10^18^) pieces of data were produced daily. The IoT data generation differed in speed, scalability, dynamics, and heterogeneity. Concerning the data rate, the challenges were due to handling data generated at high sampling rates, as they require efficient and faster processing.

Conversely, data generated at low data rates also introduced risks of losing some information. IoT device mobility added complexity because the sensing data would have to operate and reflect its performance at different locations. Product delicacy in some applications, especially when deployed in harsh environments, could affect data generation, as devices may underperform due to failure or intermittent connections. The IoT industry anticipated the creation of various applications with mixed data models and formats [1].

There are many IoT characteristics to consider in the physical layers, data generation, communication protocols, and operating modes. The understanding in this paper was that it could not be realistic to conceive a unique IoT noise-generating model that satisfied all the requirements. 

In this paper, the IoT noise generator (ING) units transmitted modulated random signals at controllable time intervals called transmit rates. On power up, the modules were idle and could broadcast no signal. When the button was pressed, the microcontroller incremented the transmit rate counter from transmit rate Tx0 to transmit rate Tx10. After Tx10, the counter overflowed and reset to Tx0. As shown in Table 1, the microcontroller set pre-programmed time intervals for individual transmit rates. An LED was programmed to flash according to the transmit rate. With this operating mode, the higher the transmit rate, the more the ING units emit RF signals into free space and the higher the amount of ambient RF activities; therefore, the system could emulate the quantity of radio noise deposited in the environment.

The design of the ING hardware followed these specifications:Three board variants: 433 MHz, 868 MHz, and 2.4 GHz;Microcontroller: Texas Instruments^®^ CC1352R;RF balun and matching network: discrete components;RF front-end: RF-specified capacitors and inductors;Printed circuit board (PCB) trace antennas: custom design of three separate models for each frequency;Battery-operated: CR2032; 3V;Push-button;LED signal indicator;Low cost;Compliance with local regulations on electronic communication [8].

Figure 3 introduces the functional block diagram of the ING. Figure 4 shows the three PCB variants. Color codes red, green, and black differentiated the 433 MHz, 868 MHz, and 2.4 GHz variants, respectively. Each PCB required a specific on-board trace antenna designed according to the frequency of interest. The experiment included 15 boards on each frequency. The production delivered a total of 45 PCBs.

## 5. Data Collection Campaign

The data collection campaign was conducted outdoors at five candidate sites in the Kwazulu-Natal province, South Africa, as shown in Table 2. Figure 5 illustrates the measurement campaign’s standard operating procedure (SOP). The USRP hardware involved arranging power, ethernet cables, and the front-end components for the frequency under test as part of the hardware preparation. Inserting the external batteries was the only task required to power up the boards. By default, the ING units started in radio silence mode, Tx0. A dedicated button press incremented the transmit rate from Tx0 to Tx10. The operation required all fifteen ING units to emit noise under the same transmit rate. The recording lasted 5 min. Figure 6 shows the outdoor setup of equipment at the candidate sites as measurements were underway. A Google^®^ Earth map, in Figure 7, illustrates the locations of interest. Easy accessibility to the venues was one of the considerations dictating the selection of these locations. As an outdoor campaign, prudence cautioned us to avoid areas where the experiment would have attracted unpleasant curiosity. As an RF operation, each ING unit emits radio signals randomly. It was necessary to provide an uninterruptible power supply (UPS) as a power source for the whole system. The GRC manipulation was to set the frequency, edit the filename, start/stop the recording, and save the raw data.

For this research, the routine of data collection produced the following:Eleven transmit rates (Tx0–Tx10);Thirty-three raw datasets (three frequency bands per site);A total of 165 raw datasets (5 candidate sites);One raw dataset weighted 30 GB in 5 min recording;Required storage: 4.95 TB;Raw data type: linear numerical;Estimated minimum operation time: 3 h per site.

## 6. Preparing Data for Analysis

### 6.1. Dealing with Big Data

The concept of big or large data is defined relative to the computer hardware and software systems employed to operate this data. Factors including the data volume, required storage, processing capability, and data format dictate the difficulty levels a machine may face to carry out data-intensive tasks. Big data often leads to computer infrastructure struggling to process and store the given data timeously. Big data is no longer valid when the machine can effortlessly process and store a volume of data [30,31]. Scientific measurements are one of the generators of big data. They are often made at a high time resolution and start to become excessive when they involve two or three dimensions of space. Dealing with big data requires additional hardware and software capabilities to help ensure seamless and fast computing [30]. Frequently, memory and CPU are the components exposing machine weaknesses. Some software operates by loading the entire volume of data in the RAM. Processing would take a long time or eventually fail as the dataset size is larger than the PC memory. Distributed computing has successfully addressed big data by providing the advantage of completing challenging tasks by parallel processing [32].

The literature reveals that the options for cloud computing, multiple parallels, and distributed computing come with a cost, time, and effort. Implementing these methods would contradict the strategy set in this paper to offer simplified and cost-effective solutions. MathWorks^®^ proposed some techniques for efficient computer memory usage [12,33,34]. Furthermore, divide-and-conquer is one of the fundamental approaches for data reduction, which consists of decomposing extensive data into small partitions, applying the required processes to individual partitions, and then recombining the individual results together [11,35]. Inspired by this, this research adopted a data-wrangling approach to transforming the raw dataset from 10 billion rows (30 GB) to a reduced dataset of 10 million rows (30 MB), suitable for computational analysis. The techniques in this approach required, in the first instance, to exclude every nil value, since the interest in this study was solely on non-zero levels. Secondly, the average values were calculated by data segmentation: splitting non-zero samples to produce 10 million segments, applying the arithmetic average on data in each segment, and merging the 10 million averaged results to bind a reduced dataset [12]. Figure 8 illustrates this data reduction method. Programmatically, Algorithm 1 is a procedural code snippet in Python to implement data reduction for DUT_433_Tx0 data using Pandas and NumPy libraries. The arithmetic mean, median, and peak characterize the radio noise. The metric used depends on the continuous or impulsive nature of the noise. The mean is convenient for Gaussian noise. One of the specifications for white Gaussian noise definition is a zero mean value. On the contrary, the median is not affected by pulses, and is therefore a good statistical metric when analyzing non-Gaussian noise. Peak values are arbitrary [28,36]. Knowing that IoT products are configured for short duration air-time (in order of milliseconds), it should justly imply that these products would generate impulsive radio noise. With that in mind, this paper employs the arithmetic average. This paper suggests that the myriad IoT radio activities continuously occupy the environment by considering the random and multiple RF transmissions, the scalability, and the number of deployable IoT units. In aggregate, the radio receiver does not experience the essence of impulsive noise.

### 6.2. Calibration and Amplitude Correction

A radio noise measurement system shall distinguish between the internally generated noise by the receiver and the external noise captured by the antenna. The collected raw data combined both variables. The internal noise measurement required inserting a 50 Ω load at the antenna connection point. Subtracting the internal noise values from the collected raw data results in obtaining the actual external noise levels [19]. This paper measured the equipment noise (EN) for five minutes after replacing the receiving antenna with one 50 Ω RF load. This procedure also applied data reduction techniques to EN [12].
**Algorithm 1.** Code snippet for data reduction method.*# 1. Import libraries*Import numpy as npimport pandas as pd*# 2. Import raw data*binary_file = ‘C:/…/…/DUT_433_Tx0’rawdata = np.fromfile(binary_file, dtype = np.float32);print(“File size:”, round(rawdata.nbytes /1e9,2), “GB”);*# 3. Remove the zeros*rawdata = rawdata[rawdata != 0];*# 4. Data segementation*number_of_chunks = int(10e6); chunks = np.array_split(rawdata, number_of_chunks);*# 5. Compute arithmetic means for each chunk*list_of_means = [np.mean(c) for c in chunks]; *# 6. Load the means in dedicated variable for further analysis.*DUT_433_Tx0 = np.array(list_of_means);

The results in this RNSS obtained lower EN levels than the antenna noise (AN) on the frequencies and candidate sites. The calibrated noise (CN) resulted from the subtraction, as follows: CN = AN − EN. The antenna factors were insignificant. The test results show that the losses due to RF cables, connectors, and RF adaptors were also negligible [12].

The LNA gain, the filter insertion loss, and the digital gain set in the URSP constituted the RNSS front-end factor. The amplitude correction process subtracted the front-end factor from the CN data. Table 3 shows the values considered for the amplitude correction [12].

### 6.3. Amplitude Scaling

The amplitude-corrected dataset contained positive and negative values. Shifting all the samples to the positive number range was essential to prevent errors with mathematical operations. The results did not change the relational proportions within the datasets. This amplitude scaling required adding to every sample $X_{n}$ the absolute value of the minimum $\left|X_{min}\right|$. To avoid conflicting errors with a mathematical process, such as logarithm on samples of zero volt amplitude, Equation (15) adds a small margin of 0.000001 ($10^{-6}$) to $\left|X_{min}\right|.$ Smith [25] advises adding a negligibly small number as an alternative to handle a mathematical nuisance with zero. Finally, Equation (15) computes radio noise relative power by squaring the amplitude-scaled data.
(15)$$P_{r}={\left\{X_{n}+\left(\left|X_{min}\right|+10^{-6}\right)\right\}}^{2}\times 10^{3}\;$$

## 7. Statistical Analysis

For a descriptive statistical analysis, this paper focuses on the measures of central tendency and measure of variability, with an interest in statistical metrics, such as mean, standard deviation, variance, kurtosis, and skewness. The kernel density estimate (KDE) function explored the observation distributions. The average, median, and mode represent measures of central tendency. The standard deviation informs of measures of variability or dispersion from the center [37]. The skewness value characterizes the shape of the distribution: it is symmetrical when the skewness is nil and has a right tail or is skewed to the right when the skewness is a positive value. Otherwise, the distribution has a left tail or is skewed to the left when the skewness is negative. A kurtosis value of zero indicates normality. Positive kurtosis relates to a peaked distribution, and negative kurtosis obtains a flatter distribution [38]. 

The analysis considered three features: the candidate sites, frequency bands, and transmit rates (from Tx0 to Tx10). This survey produced 165 sets of raw data, each requiring analysis of the mean, standard deviation, variance, kurtosis, skewness, median, first and third quartiles, interquartile range, and minimum and maximum scores of the boxplot. Table 4 and Table 5 present these metrics obtained from the NGM-868 MHz case for brevity. Table 4 demonstrates the positive and slightly high values for skewness. High kurtosis expects samples to have slightly right-tailed and peaked distributions. The standard deviations and variations were very close, indicating that the data had minimum deviations from the mean values. An analysis exploited the elements of the boxplot in Table 5 to remove the outliers. 

The initial density and boxplot graphs, not shown in this paper for brevity, demonstrate the presence of outliers in the datasets. To filter out the outliers, the process required an evaluation of the elements of the boxplots, such as the lower fence (LF) or min score, the 1st quartile at the 25th percentile, the arithmetic median, the 3rd quartile at the 75th percentile, the upper fence (UF) or max score, and the interquartile range. Further analysis manipulated only the outlier-less data contained within the LF and UF boundaries.

To analyze the statistical nature of these data, Figure 9 combines the density plots for all five sites and the respective frequency bands. To demonstrate the direction of the trends, Figure 10 shows the regression lines and scatter plots obtained from the mean values of the individual transmit rates. The *x*-axis in Figure 10 represents the transmit intervals in seconds (see Table 2).

## 8. Discussions of Results

Figure 9 and Figure 10 brought some aspects to interrogate; however, the discussions prioritized answering this paper’s question: what is the influence of IoT emissions on the radio noise environment? One of the first observations was that the mean values at the distributions’ centers were very close to each other through Tx0 to Tx10 in their respective groups (site and band). The density plots in Figure 9 confirm the elimination of the outliers, as there were no visible isolated values. Additionally, there were no observations of skewed distributions indicating near-zero skewness values. These graphs marked interesting resemblances, especially in their respective frequency bands:The 433 MHz patterns exhibited characteristics of a normal distribution with noticeable Gaussian bell-shaped curves. Of course, the mean values of each transmit rate, supposedly at the center of the distributions, showed different values; however, the graphs remained symmetrical and centered around their mean through Tx0 to Tx10. The distribution shapes within each candidate site demonstrate that the data carried minor differences in standard deviations and variances, although small.The 868 MHz distributions demonstrated some abnormalities. The MNC and MNS data showed flattened distributions at their respective Tx10 graphs, indicating negative kurtosis values; however, the same sites presented peaked distributions with lower transmit rates exhibiting positive and high kurtosis. DUT, GLM, and NGM showed quasi-normal distributions; however, DUT and GLM counted fewer samples in their Tx10 data.The 2400 MHz patterns also exhibited the same characteristics as the normal distributions; however, the data showed ripples.

The slope of the regression lines in Figure 10 helped to easily visualize the impact of increasing the transmit rates, thus multiplying radio activities in environments. Out of 15 assessments, only 3 cases (20%) showed a decreasing trend. Twelve cases (80%) indicated an augmenting direction. The levels of radio noise power in the linear scale show minimal differences, with the rates of the RF activities increasing; however, these variations, though small, may have significant merit in the context of environmental radio noise.

In a university environment, such as the DUT Steve Biko Campus, ambient radio noise increased as the rate of the RF activities increased, irrespective of the operating frequency. The minimum amplitude was 10.529603 mW, and the maximum was 10.541081 mW at 433 MHz. At 868 MHz, the minimum was 1003.463161 mW, and the maximum was 1003.604141 mW. At 2400 MHz, the minimum was 38.965016 mW, and the maximum was 38.995504 mW. In the GLM site, the ambient radio noise at 433 MHz decreased while the rate of the RF activities increased. The maximum was 2.845122 mW, and the minimum was 2.844545 mW; however, the trends were augmented for 868 MHz (minimum 82.883005 mW, maximum 82.922968 mW) and 2400 MHz (minimum 53.710115 mW, maximum 53.740330 mW). In a suburban area such as Montclair, the ambient radio noise increased as the rate of the RF activities increased through all the frequency bands, namely, at 433 MHz (minimum 2.804772 mW, maximum 2.807212 mW), 868 MHz (minimum 1031.475526 mW, maximum 1031.693036 mW), and 2400 MHz (minimum 53.710115 mW, maximum 53.740330 mW). In an urban area such as Morningside, the ambient radio noise increased with the RF activities at 433 MHz (minimum 2.846615 mW, maximum 2.924101 mW) and 868 MHz (minimum 325.305783 mW, maximum 325.350955 mW). The trend declined at 2400 MHz (maximum 70.211370 mW, minimum 70.200025 mW). In an industrial environment such as New Germany, the ambient radio noise was increased with the RF activities at 433 MHz (minimum 2.591371 mW, maximum 2.592906 mW) and 868 MHz (minimum 589.076361 mW, maximum 589.092486 mW); however, the trend declined at 2400 MHz (maximum 186.602416 mW, minimum 186.409612 mW).

## 9. Empirical Analysis

This heuristic and empirical method was another way to test the influence of IoT radio operations on environments. The test analyzed the mean values obtained at different transmit rates for each candidate site and frequency. The exercise consisted of identifying the features or transmit rate columns containing the highest average value per site and frequency and then splitting these features into two categories after applying the “50 percent plus one” rule:Lower transmission rate side (LTRS): grouping transmit rates Tx0 to Tx5;Upper transmission rate side (UTRS): grouping transmit rates Tx6 to Tx10.

Table 6 shows the transmit rates of the maximum noise and counts how many fell under LTRS or UTRS. The analysis demonstrates that the UTRS category dominated and therefore confirmed the theory that the level of ambient radio noise is directly proportional to the volume of RF activities, even in this case of low-power IoT operations, irrespective of the frequency bands and type of environment (industrial, urban, or suburban).

## 10. Conclusions

This study has developed a simple and cost-effective instrument to collect radio noise environment. The radio noise surveying system (RNSS), built with software-defined radio, demonstrated excellent performance with the equipment noise (EN) being lower than the level captured by the antenna. The results confirm that the RNSS could collect and record ambient radio noise. The system setup and standard operation procedure were quickly repeatable at different candidate sites. While the IoT industry was still not fully deployed, the research innovated a method to generate and simulate radio noise. This paper proposes a strategy to convert big data to a reduced dataset size by removing the zeros and averaging the data by segmentation, resulting in final datasets ready for smooth computing. The exercise handled 165 datasets. The algorithms designed for data analysis demonstrated agility and performance replicable to all datasets featuring the candidate sites, frequency bands, and transmit rates, providing informative results for this study.

The slopes of the regression lines demonstrate that 80% of the analyzed data developed augmenting trends. Only 20% of the data showed declining trends. Both statistical and empirical analyses concluded with the same observation that the exponential growth of the presence of IoT will generally cause the ambient radio noise to increase irrespective of the candidate sites in urban, suburban, or industrial environments, and irrespective of the operating frequency. Even with low RF power, the IoT applications could still contribute to crowding the radio spectrum and, therefore, could pause risks of limited performance and sustainability when fully deployed at a large scale. This revelation is relevant to science in general, the engineering in the electronic industry, and the regulatory authorities. This study opens up opportunities for future recommendations: (1) recorded data should train machine learning models to predict ambient radio noise levels; (2) the RNSS demonstrated performance repeatability. The same tool should extend the measurement campaign to survey many more candidate sites to provide a broader representative map; (3) this paper limited itself to three frequencies (433 MHz, 868 MHz, and 2.4 GHz). The same routine should be employed to examine radio noise beyond these bands; (4) as this paper allowed random survey times, other research should try to characterize the radio noise environments at specific times, under different weather conditions, and in different seasons.

## Figures and Tables

**Figure 1 sensors-23-07899-f001:**
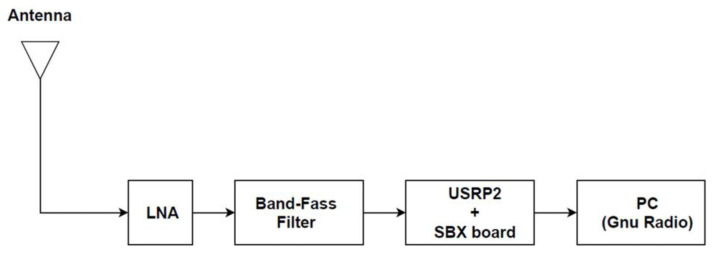
RNSS signal chain.

**Figure 2 sensors-23-07899-f002:**
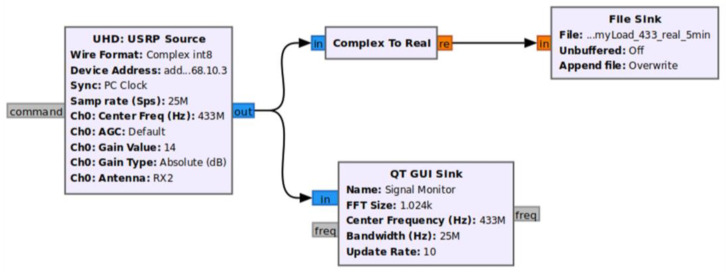
GNU Radio receiver configuration.

**Figure 3 sensors-23-07899-f003:**
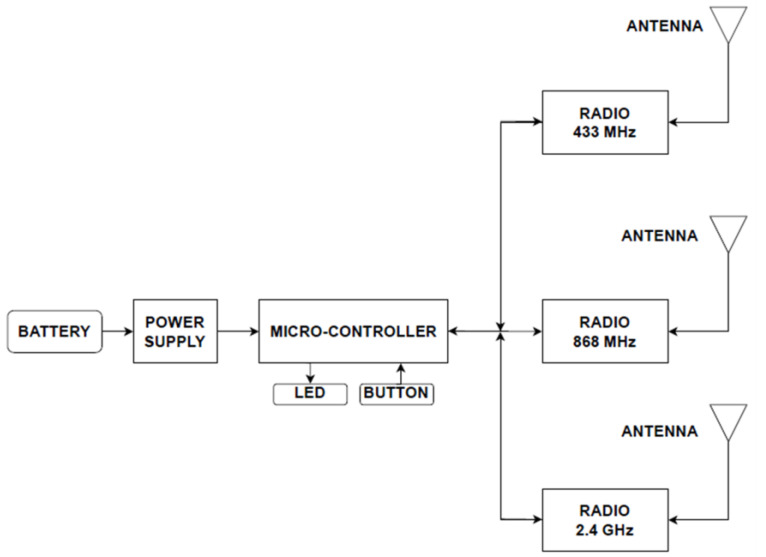
Functional schematic of the IoT noise generator hardware.

**Figure 4 sensors-23-07899-f004:**
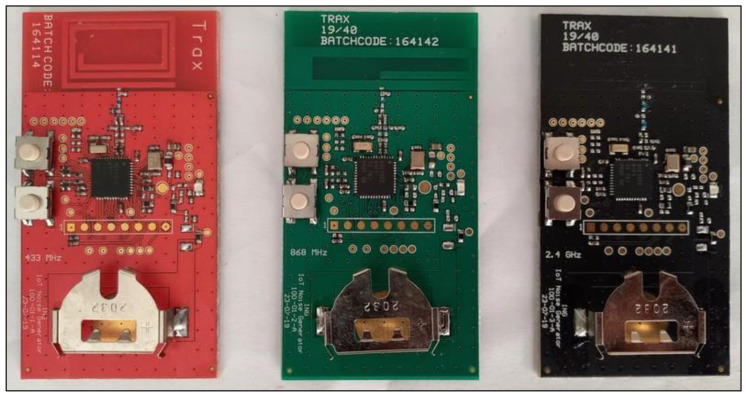
Assembled boards of the IoT noise generators.

**Figure 5 sensors-23-07899-f005:**
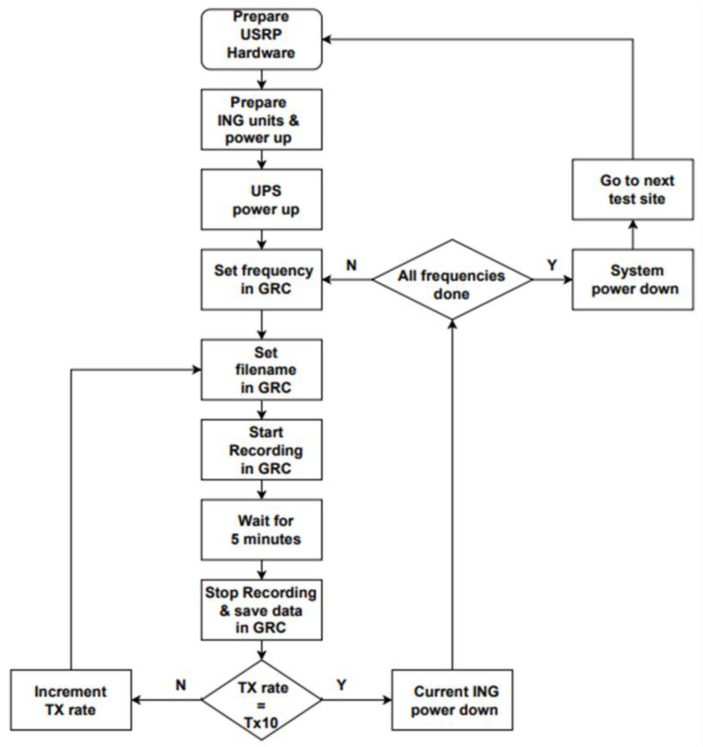
Measurement campaign SOP.

**Figure 6 sensors-23-07899-f006:**
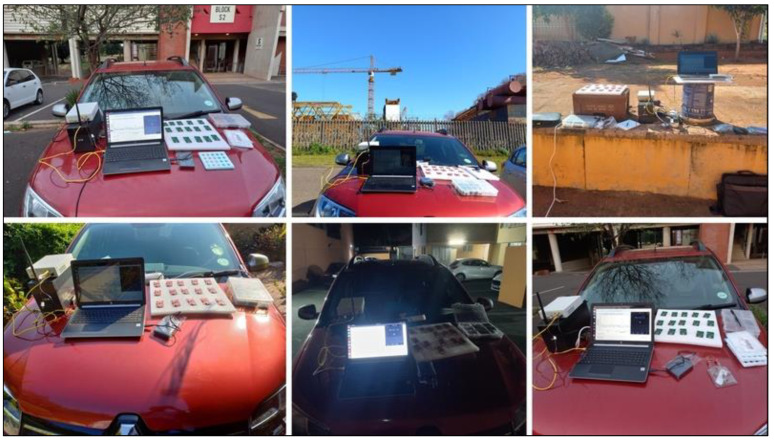
Equipment setup during the surveying campaign.

**Figure 7 sensors-23-07899-f007:**
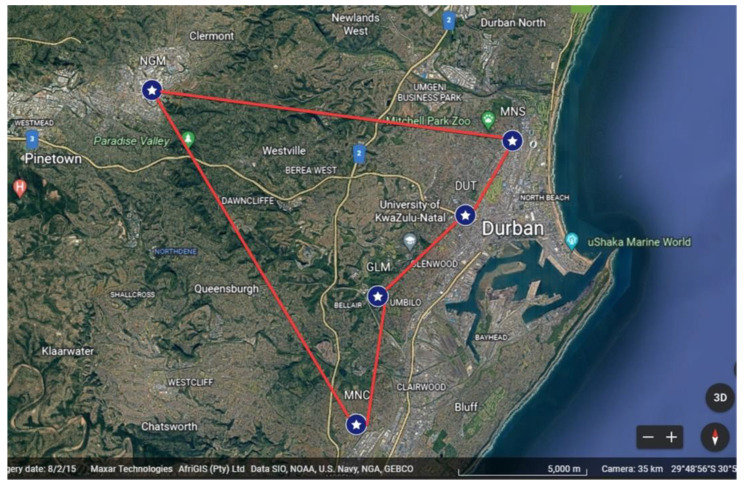
Google Earth map of candidate sites.

**Figure 8 sensors-23-07899-f008:**
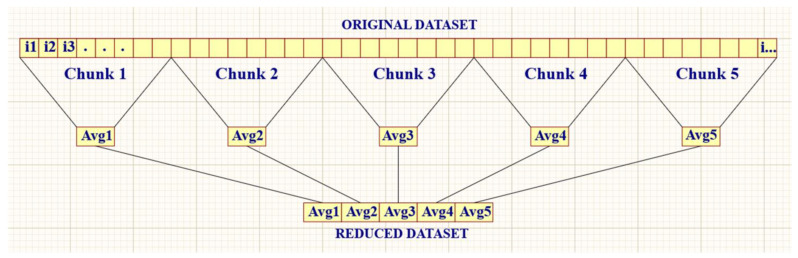
Method of data reduction.

**Figure 9 sensors-23-07899-f009:**
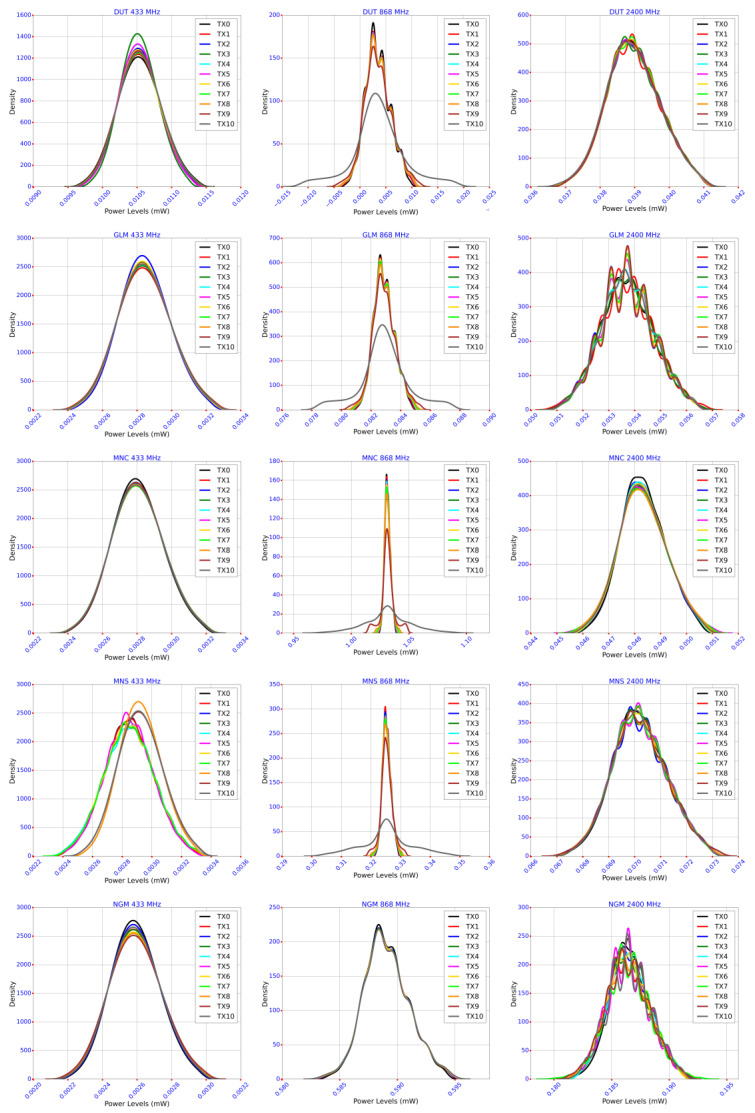
Density plots grouped per candidate sites and frequency bands.

**Figure 10 sensors-23-07899-f010:**
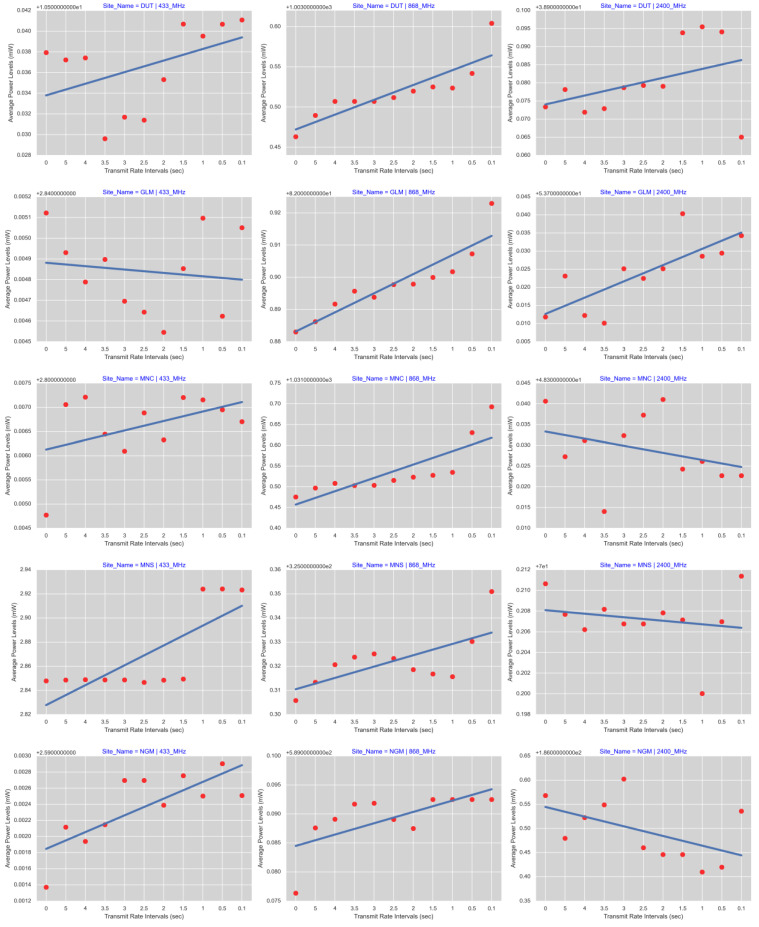
Regression lines and scatter plots grouped per candidate sites and frequency bands.

**Table 1 sensors-23-07899-t001:** Transmit rates and corresponding time intervals.

Transmit Rate	Transmit Interval (Seconds)
Tx0	Idle
Tx1	5
Tx2	4
Tx3	3.5
Tx4	3
Tx5	2.5
Tx6	2
Tx7	1.5
Tx8	1
Tx9	0.5
Tx10	0.1

**Table 2 sensors-23-07899-t002:** Candidate sites for IoT noise surveying campaign.

Site	Description	Date
Berea Steve Biko Campus (DUT)	Urban	6 July 2022–8 July 2022
Glenmore (GLM)	Suburban	10 July 2022
Montclair (MNC)	Suburban	16 July 2022
Morningside (MNS)	Urban	19 August 2022
New Germany (NGM)	Industrial	30 June 2022–4 July 2022

**Table 3 sensors-23-07899-t003:** Amplitude correction values.

Band	Front-End Factor	USRP Gain	Amplitude Correction
433 MHz	21.3 dB	14 dB	7.3 dB
868 MHz	16.2 dB	14 dB	2.2 dB
2.4 GHz	8.1 dB	14 dB	−5.9 dB

**Table 4 sensors-23-07899-t004:** Statistical characteristics of IoT radio noise at NGM-868 MHz.

Site	Band	Stat.	Tx0	Tx1	Tx2	Tx3	Tx4	Tx5	Tx6	Tx7	Tx8	Tx9	Tx10
NGM	868 MHz	Mean	0.589115	0.589247	0.589307	0.589345	0.589376	0.589327	0.589551	0.589254	0.589254	0.589254	0.589254
		Std	0.002647	0.017835	0.021365	0.023126	0.024879	0.022509	0.032495	0.017900	0.017900	0.017900	0.017900
		Var	0.000007	0.000318	0.000456	0.000535	0.000619	0.000507	0.001056	0.000320	0.000320	0.000320	0.000320
		Kurt	56.083614	355.496605	265.715441	274.107725	222.432388	856.255615	172.717403	142.075992	142.075992	142.075992	142.075992
		Skew	−0.109029	7.155354	6.436797	7.073440	6.198063	14.221488	5.956075	3.099585	3.099585	3.099585	3.099585

**Table 5 sensors-23-07899-t005:** Elements of boxplot for IoT radio noise at NGM-868 MHz.

Site	Band	Stat.	Tx0	Tx1	Tx2	Tx3	Tx4	Tx5	Tx6	Tx7	Tx8	Tx9	Tx10
NGM	868 MHz	Min Score	0.583820	0.583504	0.583466	0.583315	0.583164	0.583350	0.583158	0.583123	0.583123	0.583123	0.583123
		Q1	0.587752	0.587669	0.587688	0.587645	0.587606	0.587673	0.587602	0.587601	0.587601	0.587601	0.587601
		Median	0.588895	0.588944	0.588952	0.588908	0.588915	0.588957	0.588923	0.588889	0.588889	0.588889	0.588889
		Q3	0.590374	0.590446	0.590503	0.590532	0.590568	0.590555	0.590564	0.590587	0.590587	0.590587	0.590587
		IQR	0.002622	0.002777	0.002815	0.002887	0.002962	0.002882	0.002963	0.002986	0.002986	0.002986	0.002986
		Max Score	0.594306	0.594611	0.594726	0.594861	0.595011	0.594878	0.595008	0.595066	0.595066	0.595066	0.595066

**Table 6 sensors-23-07899-t006:** Transmit rates of the highest average level of IoT noise.

Band	DUT	GLM	MNC	MNS	NGM	LTRS	UTRS
433 MHz	Tx10	Tx1	Tx2	Tx9	Tx9	2	3
868 MHz	Tx10	Tx10	Tx10	Tx10	Tx6	0	5
2.4 GHz	Tx8	Tx9	Tx10	Tx10	Tx4	1	4

## Data Availability

Raw data and processed data are available on request.

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
