# Peer review of "An Assessment of Environmental RF Noise Due to IoT Deployment"

_sensors, 2023, doi:10.3390/s23187899_

Round 1

Reviewer 1 Report

This paper presents the development of a cost-effective Radio Noise Surveying System (RNSS) using Software Defined Radio equipment. The RNSS demonstrated superior performance, as its equipment noise was lower than the ambient noise captured, ensuring accurate recordings. The system is easily replicable across various sites. The manuscript is clear, relevant to the field, presented well-structured, and scientifically sound. The manuscript’s results are reproducible based on the details given in the methods section. However, all figures with charts could be bigger because they couldn’t be read.  Also, I think the paper needs in conclusion to mention more about their future work.  

Minor English edits

Author Response

1

The figure size could be bigger

Figures resized.

Some images are high-resolution (300 dpi). They should be readable even if printed.

2

Mention future work in the Conclusion

Addressed in 556 - 563

3

English language editing

Reviewed by professional editing services.

Reviewer 2 Report

The paper describes the performed research by the authors to do an assessment of environmental RF noise due to IoT deployment. 

The paper is well written and structured. 

Several issues that this Reviewer has found are summarized below:

1. Abstract Section. The abstract section should be enhanced at the end of its part by adding 2-3 propositions discussing the main outcome of the obtained experimental results. 

2. Introduction Section. At the last part of the introduction, prior to the paper structure description, the Authors have to add a small paragraph (2-3 sentences) describing the aim and novelty of the research contribution.

3.  The following sentences are too vague. More explanation is needed. "The PCB trace antennas were simulated and tested. This exercise consisted of fifteen ING boards per board variant." 

What does it mean fifteen ING boards per board variant ?

Over remark: English should be reviewed by a professional English Speaker.

Author Response

1

Add experimental results in the abstract.

Addressed in 17 - 18

2

Describe the aim and novelty of the research

Addressed in 50 – 59, 63 -64, and 68 - 74

3

Clarify the sentence “PCB trace antennas…” in 206 - 207

Addressed in 299 - 301

4

English language editing

Reviewed by professional editing services.

Reviewer 3 Report

This manuscript analyses the effects the radio emissions produced by IoT products would have on the ambient levels of radio noise within the unlicensed frequency bands of 433 MHz, 868 MHz, and 2.4 11 GHz. A simplified Radio Noise Surveying System (RNSS) was deployed for data collection of ambient radio noise from five South African candidate sites. The manuscript is interesting and well written. My specific comments are as under:

·       At the end of section 1, provide the novelty of your work. How is your work different from the work existing in the literature?

·       The literature review is very short and the authors are encouraged to conduct and report an extensive literature review.

·       Any specific reason for selecting the sites given in Figure 7?

·       Which average is used in Figure 8? If the Arithmetic mean is used, justify its usage. Especially, if there are extreme values, the Median can be a very important average.

·       Explain Figure 9 and Figure 10.

·       Please add the study limitations and future recommendations to the conclusion section.

good

Author Response

1

Provide novelty at the end of section 1

Addressed in 50 – 59, 63 -64, and 68 - 74

2

The literature review is too short

Addressed in Section 2

3

Provide reasons for selecting these sites in Figure 7.

Addressed in 317 - 320

4

In Figure 8, if the average was the arithmetic mean, then justify it. The median can be important in the case of extreme values.

Addressed in 373 - 386

5

Explain Figure 9 and Figure 10

Addressed in section 8

6

In Conclusion, add limitations and future recommendations

Addressed in 556 - 563

7

English language editing

Reviewed by professional editing services.

Reviewer 4 Report

the references need to be updated, you should write the references to explain why you chose these three frequencies 

more work need to manipulate the data 

more detailed should be added to explain the structure 

Comparison table should be added to explain your idea 

the Organization of the paper should be improved 

Author Response

1

Justify the selection of the three frequencies with references.

Addressed in 50 - 59

2

More work is needed to manipulate the data.

 Addressed in 435 – 442

3

More details are needed to explain the structure.

Addressed in 79 - 88

4

A comparison table should explain your idea 

Addressed in Tables 4 and 5

5

English language editing

Reviewed by professional editing services.

Reviewer 5 Report

The proposal is not properly justified for publication, it is recommended to improve the following:

The authors indicate a contribution from a strategy, 54-55, but reading the document does not present an adequate development of this proposal in the development of the article, it is recommended to indicate with a very specific section the contribution made by the authors.

It is recommended to include a brief explanation regarding the limitation of the transmission bands to be analyzed in the study.

Indicate in a more specific way in the proposed article why it is necessary to use GNU Radio software as a complement to the analysis.

With respect to the reference [20], what is the essential difference proposed by the authors?

Is it not indicated why it is necessary to work only on the real part of the information?

Preparation of data for analysis, is the main contribution of the article and I consider that it is not properly developed, it should be expanded in terms of a theoretical analysis and results that validate each of the steps analyzed.

The authors do not indicate the basic analytical procedure to understand and validate the proposal, this part is essential for the objective of the article.
The explanation they give, 264-269, is very simple and does not allow validation that valuable information is not lost.

It is confusing how they define CN, as the authors state it is an instantaneous measurement of antenna noise, please revise and clarify.

The Conclusions need to be re-written according to what was observed.

Minor editing of English language required

Author Response

1

Contribution mentioned in 54-54, but the document doesn’t present adequate development of this contribution.

Addressed in 373 – 385, and Algorithm 1

2

Indicate contribution with a specific section.

Addressed in 50 – 59, 63 -64, and 68 - 74

3

Explanation regarding the limitation of the transmission bands

Addressed in 50 - 59

4

Why it is necessary to use GNU Radio software as a complement to the analysis.

Addressed in 138 - 140

5

With respect to the reference [20], what is the essential difference

Addressed in 154 - 158

6

Why it is necessary to work only on the real part of the information?

Addressed in 172 - 247

7

Preparation of data for analysis should be expanded in terms of a theoretical analysis.

Addressed in 172 – 247,  373 - 380

8

The authors do not indicate the basic analytical procedure to understand and validate the proposal.

The explanation they give, 264-269, is very simple and does not allow validation that valuable information is not lost.

Addressed 363 – 365

9

It is confusing how they define CN, as the authors state it is an instantaneous measurement of antenna noise, please revise and clarify.

I don’t find this remarque in the paper.

Section 6.2 is the only place discussing CN. I can’t find this statement.

10

The Conclusions need to be re-written according to what was observed.

Addressed in section 10

11

English language editing

Reviewed by professional editing services.

Round 2

Reviewer 3 Report

As the authors addressed my concerns, I recommend the paper for publication in its present form.

Fine

Reviewer 5 Report

The new proposal is more suitable for your reading and understanding of what the authors have done.